# Mechanisms of Moral Disengagement in the Transition from Cybergossip to Cyberaggression: A Longitudinal Study

**DOI:** 10.3390/ijerph18031000

**Published:** 2021-01-23

**Authors:** Daniel Falla, Rosario Ortega-Ruiz, Eva M. Romera

**Affiliations:** Department of Psychology, Universidad de Córdoba, 14014 Córdoba, Spain; daniel.falla@uco.es (D.F.); ed1orrur@uco.es (R.O.-R.)

**Keywords:** cybergossip, cyberaggression, moral disengagement, cognitive restructuring, longitudinal design

## Abstract

The internet is an area where young people establish relationships and develop socially, emotionally and morally, but it also gives rise to certain forms of online behaviour, such as cybergossip, which are associated with cyberaggression and other risky behaviour. The aims of this study were to verify whether a longitudinal association exists between cybergossip and cyberaggression, and to discover which mechanisms of moral disengagement may mediate this relationship. The final sample consisted of 1392 students (50% girls; *M*_age_ = 13.47; SD = 0.77), who were surveyed in a three-wave longitudinal study at six-month intervals. The results obtained confirmed a direct, positive relationship between cybergossip, subsequent cyberaggression and the mediation exerted by cognitive restructuring in this transition. We discuss the importance of recognizing and detecting the fine distinction between online gossip and cyberaggression with the intention of doing harm, and focus on the justifications used by young people to normalize online bullying. To sum up, there is a clear need to encourage ethical, responsible behaviour in online interactions in order to achieve well-balanced, more sustainable relationships in classrooms.

## 1. Introduction

The recent rise in access to digital devices and the significant increase in online communication mean that many aspects of young people’s interpersonal relationships are carried out on social networks. In these ephemeral channels of social media, all the important psychosocial processes, which occur in young people, and particularly in the unstable years of adolescence, take place. When this communication is conducted within reasonable levels of decency and consideration for others, the use of social networks and in general, the communication between peers through these channels, seem to foster positive values of social support and emotional balance (see meta-analysis [1]). In fact, online behaviour and communication, if not marred by other problems, has a huge potential to widen the social horizons of both boys and girls [2].

However, one aspect of online behaviour which young people participate is cybergossip. It is defined as an online behaviour which implies sharing comments (positive, negative or neutral) with others about a known person who is not present. [3]. There is no general consensus in the studies on gossip and cybergossip that this phenomenon is necessarily a worrying one. In fact, some studies, well supported by objective data, claim that it can be a useful way of consolidating the norms of the group and boosting the social standing of its members [4], as well as providing an entertaining way of strengthening existing relationships and creating new bonds [5]. However, it is also true that gossip, both face-to-face and online, can have negative connotations associated with the idea of manipulating or influencing the group’s feelings about one of its members, which could result in cyberaggression and cybervictimization when there is an intention to harm [6]. Despite this, no studies, to date, have explored the question of whether there is a longitudinal relationship between cybergossip and cyberaggression. In addition, online messages may be more susceptible to misinterpretation, without the support of non-verbal communication [7]. Although initial studies found that girls were involved more often in this type of online behaviour than boys [8], more recent studies show that cybergossip is equally common among boys and girls [9].

### 1.1. Cyberaggression and the Immorality of Bullying

Cyberbullying is an example of immoral behaviour, which damages both the socialization process and an individual’s moral development and sensitivity, which that must be built throughout the years of childhood and adolescence [10]. Young people take advantage of the possibilities offered by Information and Communication Technologies (ICT) to engage in behaviour such as intimidation, harassment, mockery, attacks on social prestige and, in general, aggressive behaviour towards their peers, which weakens the self-esteem and confidence of the victims of their attacks [11]. Such behaviour and attitudes are without doubt immoral, because they break the convention of ethical reciprocity which peers need to relate to each other [12] and contribute to an imbalance of social power, which makes cybervictimization worse [13]. In addition, cyberbullying makes all the damage of the intentionality and the imbalance of power worse by adding the factors of anonymity and the fact that the audience is now present 24/7 [14]. A recent systematic review of longitudinal studies revealed a prevalence of cyberaggression ranging between 5.3% and 66.2% [15], which tells us, on the one hand, that its variability is extremely wide, and on the other, that such behaviour may be very widespread. Moreover, boys and girls are engaging in this behaviour at an increasingly earlier age [16], but the data show that adolescence is the stage where these online attacks are most common, particularly the early adolescent years [11], and that boys are initially those who do it the most, although the balance tends to even out with age [17].

Descriptive studies show that the commonest form of cyberaggression developed by young people on social media consists of posting rumours or gossip with the intention of doing harm [18]. Therefore, in one latent classes study, a specific profile was found, which was referred to by the authors as ‘gossiping perpetrator-victims’ [19]. This involved acting as aggressor and victim at the same time within the framework of gossip. However, despite the fact that cross-sectional studies have found than, but they are two different phenomena, there is a direct and positive relationship between cybergossip and cyberaggression [9], no specific longitudinal studies have been performed to discover whether there is a longitudinal relationship between these two aspects of online behaviour. Similarly, there have been no studies of variables that may be associated with cybergossip and that could mediate between both types of online behaviour. Presumably, the factor which turns simple ‘gossip’ into aggressive social behaviour, or in other words immoral behaviour, is the intentionality to do harm. Therefore, open communication in normal gossip games, if conducted in bad faith or there is intent to do harm, could easily turn into cyberaggression.

### 1.2. Moral Disengagement and Its Mediating Influence on Cyberaggression

A large number of studies have shown that moral disengagement (MD) can exert an influence in cyberbullying [20]. In a previous meta-analysis, it was found to be one of the highest risk factors for the perpetration of cyberbullying (r = 0.28) [21]. MD is a set of cognitive strategies and trains of thought which allows individuals to self-justify their own antisocial and immoral behaviour when they violate ethical conventions and standards, and remove any emotional and moral reservations they may have about breaking these standards [22]. The justification for this immoral behaviour, in full knowledge that it breaks the peer code of reciprocity and respect, minimizes or removes feelings of guilt through different cognitive mechanisms, such as sharing or transferring responsibility, distorting the consequences by misrepresenting or ignoring them, or degrading the victims by blaming or dehumanizing them. These strategies of erroneous reasoning are clustered in four: cognitive restructuring, minimizing responsibility, distorting consequences and dehumanizing victims [23]. It has been shown that, despite the fact that the perpetrators may be aware of the immorality of such aggressive behaviour [24], these mechanisms of cognitive self-deception help reduce their feelings of self-reproach when perpetrating of unfairly aggressive behaviour, and remove any inconvenient feelings of remorse [25]. However, so far, no one has proved whether these mechanisms of moral disengagement take place in cybergossip, despite the fact that both occur online and both share a positive relationship with cyberaggression.

While the role of MD in traditional (face-to-face) bullying has been studied in detail, few studies have addressed moral disengagement in online behaviour [26]. Furthermore, most of the works on MD have treated this phenomenon as a one-dimensional construct [27]. However, analysing the activation of these mechanisms separately may contribute to a more accurate understanding of their role in the subtle nuances of behaviour which occur in cybergossip. In one study in children between seven and nine years old, an association was found between advantageous comparison and euphemistic labelling, both within the domain of cognitive restructuring and in the attribution of blame [28]. In another work, a relationship was found between the spread of responsibility and the attribution of blame in students between 12 and 15 years old [29]. Meanwhile, in a more recent cross-sectional study, it was observed that cyberaggression was fuelled by cognitive restructuring, distortion of consequences and dehumanization [30].

In recent years, a growing number of studies have looked into MD as a mediator in explanatory models of cyberaggression and the influence of morality on cyberaggression behaviour [21]. For example, the mediating effect of MD on the relationship between cyberaggression and personality characteristics has been tested [31,32,33], in analyses of the impact of negative family situations such as abuse or intra-parental conflict [34,35], or in the school context [36], and in all these studies it was found that MD has a mediating effect between these variables and cyberaggression. However, in most of these works, DM was used as a one-dimensional construct, ignoring the fact that it is by nature multidimensional, and its mechanisms are divided into several subtle influences, all of which are considered relevant when analysing the mediation of MD mechanisms in the transition from cybergossip to cyberaggression. This is the main subject of this work.

### 1.3. The Present Study

Both cybergossip and cyberaggression are aspects of online behaviours which occur on the Internet, especially in the early stages of secondary school [11], a period of adolescence which is associated with greater involvement in risky behaviour [37]. Previous studies have pointed out the strong relationship between cyberaggression and cyberaggression in cross-sectional studies [9], but no longitudinal studies have been performed. The immoral nature of cyberaggression has been specifically linked, in previous studies, to the use of cognitive MD mechanisms [30]. Some of these mechanisms, such as cognitive restructuring, distortion of the consequences and dehumanization of the victim could be mediating mechanisms in the transition from cybergossip to cyberaggression since its effect has been proven in cyberaggression [30]. In brief, the aims of this study are as follows: (1) to examine the longitudinal relationship between cybergossip and cyberaggression; and (2) to verify the mediating effect of the different mechanisms of MD on the relationship between cybergossip and cyberaggression. To achieve this, we will follow model 4 [38] (see Figure 1) and the following hypotheses:

**H1.** 
*Cybergossip at Time 1 will be directly and positively related to cyberaggression at Time 3, according to previous cross-sectional studies that found this association [3,9].*


**H2.** 
*Based on studies about the mediation role of moral disengagement strategies [21,30], it was hypothesized that cognitive restructuring, distortion of consequences and dehumanization (T2) will be the mechanisms that mediate the relationship between cybergossip (T1) and cyberaggression (T3).*


## 2. Materials and Methods

### 2.1. Participants

The schools were selected through non-probability sampling for accessibility [39]. In total, there were thirteen educational schools (eight rural and five urban) from the South of Spain, two of which were subsidized and eleven public. This longitudinal study included three-wave across 12 months. The first wave took place in October 2017 (T1) where 1912 students from grade 7 and grade 8 were recruited (47.5% girls; *M*_age_ = 12.66; *SD* = 0.86). In May 2018 (T2), 1625 adolescents from the same grade answered, the recruitment rate was 84.99%, being (48.7% girls; *M*_age_ = 13.07; *SD* = 0.86). The third wave (T3) was in October 2018 and had 1392 students (50% girls; *M*_age_ = 13.47; *SD* = 0.77) from grade 7 to grade 9, the recruitment rate being 72.80% compared to the first wave. The dropout in the study was due to the absence of the students on the day of the administration of the survey or due to a change of center. A logistic regression was performed to check whether the longitudinal analytical sample was representative of the total sample. There were no significant differences (all *p*s > 0.05) in the study variables between the adolescents who participated in the different time waves. According to Fritz and Mackinnon criteria [40], the sample size of the current study was suitable for detecting medium and large mediated effects.

### 2.2. Instruments

The cybergossip was measured through *Cybergossip Questionnaire-Adolescents* (CGQ-A) [9]. The instrument is made up of nine items (e.g., “I talk to my group of friends from Social Networks or WhatsApp about what happens to other classmates to have fun”) with Likert-type response options, ranging from 0 = *never*, 1 = *almost*, 2 = *normally*, 3 = *almost always*, 4 = *always*. The scale showed good validity and reliability indices in the validation study [9]. Acceptable reliability values were found in this work (ωT1 = 0.78, ωT2 = 0.83, ωT3 = 0.81). The values presented to measure the factorial structure were optimal, χ^2^ S-B = 132.21; *df* = 27; *p* < 0.001; NNFI = 0.97; CFI = 0.98; RMSEA = 0.055 (90% CI = [0.045, 0.064]).

Cyberaggression was measured through the *European Cyberbullying Intervention Project Questionnaire*, ECIPQ [41]. This instrument is made up of two dimensions of 22 items that measure cybervictimization and cyberaggression. For this study, only the second dimension was used through 11 items (e.g., “I threatened someone through texts or online messages”) on a Likert-type scale with five response options ranging from 0 = *never*, 1 = *once or twice*, 2 = *once or twice a month*, 3 = *about once a week*, 4 = *more than once a week*. The scale showed good validity and reliability indices in the original version [38]. For this study, the cyberaggression dimension obtained a good reliability (ωT1 = 0.82, ωT2 = 0.89, ωT3 = 0.90). The factorial structure with this sample was confirmed with optimal values, χ^2^ S-B = 142.03; *df* = 44; *p* < 0.001; NNFI = 0.99; CFI = 0.99; RMSEA = 0.042 (90% CI = [0.034, 0.049]).

Moral disengagement mechanisms were measured using the *Moral Disengagement Scale* [42]. The version used was that of adolescents, which consists of 24 items with five Likert-type response options and whose values range from 1 to 5, being (1 = *strongly disagree*; 2 = *slightly agree*; 3 = *strongly agree*; 4 = *strongly agree*; 5 = *strongly agree*). This scale is grouped into four strategies: cognitive restructuring (9 items), e.g., “It is okay to use force against a partner who insults your family”; minimization of responsibility (6 items), e.g., “If people are living in poor conditions they are not responsible for attacking”; distortion of consequences (3 items), e.g., “Making fun of a partner is not really hurting them”; and dehumanization (6 items), e.g., “There is nothing wrong with treating someone who behaves despicably. The four-factor structure was confirmed in other studies [27]. In the present sample, this structure was reaffirmed with four domains through a confirmatory factor analysis CFA, χ^2^ S − B = 732.48; *df* = 246; *p* < 0.001; NNFI = 0.97; CFI = 0.98; RMSEA = 0.041 (90% CI = [0.037, 0.044]). Reliability was good for cognitive restructuring at all three times (ωT1 = 0.82, ωT2 = 0.84, ωT3 = 0.85) and for minimizing responsibility (ωT1 = 0.71, ωT2 = 0.71, ωT3 = 0.74); while it was acceptable for the distortion of the consequences in the three waves (ωT1 = 0.58, ωT2 = 0.64, ωT3 = 0.62) and for dehumanization (ωT1 = 0.70, ωT2 = 0.76, ωT3 = 0.77).

### 2.3. Procedure

The Ethical Committee for Bioethics and Biosafety of the University of Cordoba approved this study, which preserved the principles set forth in the Declaration of Helsinki, on confidentiality, privacy and informed consent in the protocol established for the administration of the questionnaires. The selected centers were provided with information on the object of study and before starting the field work, they had to confirm the approval of the school board and present the written consent of the families or legal guardians of the participants.

The administration of the questionnaires was done on paper and supervised by the study researchers in the ordinary classroom. Schoolchildren were informed of the personal and private nature of the test, as well as that the answers were to be treated anonymously and confidentially. To ensure this, the questionnaires were registered with an alphanumeric code. This code was registered by students in each wave. The response time was approximately 30 min. The students who did not participate remained in the classroom, although the questionnaire was not distributed to them. The second and third administration was performed six months and one year later respectively, following the same procedure explained above. The stipulated time for data collection in each wave was three weeks.

### 2.4. Data Analysis

Through the statistical package IBM SPSS Statistics 26 (IBM, Armonk, NY, USA), the mean and standard deviation were performed for all the variables studied and in the three analyzed times; the existence of differences by gender was verified through the t-Student test, calculating Cohen’s d to obtain the effect size; and the Pearson correlation coefficient was calculated.

Subsequently, a mediation analysis was performed with the PROCESS v3.4 macro (SPSS Inc., Chicago, IL, USA), which uses least squares regression to estimate the importance and size of the direct and indirect effects in the mediation models [38]. All variables were standardized to produce standardized regression coefficients (beta) and to be able to make comparisons between the effects of each variable. For hypotheses 1 and 2, Model 4 was used where the mediating effect of cognitive restructuring T2 (first mediator), minimization of guilt T2 (second mediator), distortion of consequences T2 (third mediator) and dehumanization T2 were evaluated (fourth mediator) in the relationship between cybergossip T1 (predictor variable) and cybergression T3 (dependent variable). To control the effect of the variables gender, age, and cyberaggression T1 were included as covariates. Four-step procedure was followed to test the mediation effect [43]. Subsequently, the Bootstrapping method was used to infer the indirect effects, after generating an empirical representation of the sampling distribution of the indirect effects. In this method, a confidence interval is considered significant if it does not include zero. The relationship between the indirect effect and the total effect (PM) indicates the measure of the mediation effect, while the total effect reports the relationship between the independent and dependent variable [44].

## 3. Results

Table 1 shows the means, standard deviations, Student’s *t*-test and effect size in the three-waves. The differences by gender are observed between all the variables, except in cybergossip at time 1 and cyberaggression at time 3. The scores were higher in boys than in girls, except in cybergossip at times 2 and 3. The effect sizes presented a range between 0.12 and 0.52 with a significance level of 0.001.

At time 1, significant and positive correlations were observed between all variables, except the relationship between minimizing responsibility and age. Values ranged from 0.10 to 0.68. At time 2, no relationship was found between minimizing responsibility and age for the rest of the variables and for all those used at time 3, a positive relationship was found, the values ranged between 0.07 to 0.71 (see Appendix A). For hypothesis 1, the mediation analysis indicated, in the first step, that the total effect of the cybergossip at time one was significantly related to the cybergression at time three (β = 0.26, *t* = 9.20, *p* < 0.001) (see model 1 in Table 2). Hypothesis one was confirmed.

For hypothesis 2, in the second step, the cybergossip at time 1 was significantly associated with the four strategies of moral disconnection at time 2: cognitive restructuring (β = 0.24, *t* = 7.25, *p* < 0.001), minimization of the responsibility (β = 0.29, *t* = 8.05, *p* < 0.001), distortion of consequences (β = 0.22, *t* = 6.77, *p* < 0.001) and dehumanization (β = 0.25, *t* = 6.99, *p* < 0.001) (see model 2 of Table 2). While, in the third step, when we controlled the cybergossip, cognitive restructuring at time two was the only mechanism of moral disconnection that was significantly associated with cyber aggression at time three (β = 0.15, *t* = 3.31, *p* < 0.01), while the direct effect of cybergossip at time 1 on cyberaggression at time 3 was significant (β = 0.09, *t* = 2.63, *p* < 0.001) (see model 3 of Table 2). Figure 2 shows the significance in each of the model pathways.

Finally, in the fourth step, the percentile bootstrap method with bias correction was used, which indicated a positive association of the indirect effect of the cybergossip at time one on the cyberaggression at time three, solely through the path of cognitive restructuring (β = 0. 03, 95% CI = [0.01, 0.07]). Therefore, the mediation of this mechanism represented 25.96% of the total effect. Hypothesis two was partially confirmed since the only mechanism of moral disconnection that had a partial effect on the relationship was cognitive restructuring.

## 4. Discussion

The main objective of this work was to find out if there was an association between cybergossip and cyberaggression, over a time interval of two to twelve months, and whether this association was mediated by the mechanisms of moral disengagement (MD) in an intermediate timeframe. Our hypothesis is supported by previous studies with non-longitudinal designs [9], which have already supported this association. The immoral nature of aggressive online behaviour and the relationship between this and the mechanisms of MD support the hypothesis that cognitive disengagement strategies could act as mediators of the transition between these two aspects of online communication among peers. Cognitive restructuring, dehumanization attributed by the aggressor to the victim and distortion of the consequences could be transition processes from one phenomenon to another, as some studies have already indicated [30].

The first hypothesis was confirmed. The longitudinal analysis found that cybergossip has a direct relationship with subsequent cyberaggression behaviour, and that the cognitive disengagement mechanisms mentioned above (restructuring, dehumanization and distortion of consequences) act as a transition from cybergossip to cyberaggression. The comments and judgements about others which young people post on internet can be interpreted by those involved with different emotional and moral shades of meaning, which leads to a complex, ambiguous information network, where there seems to be a fine line between harmless gossip and harmful bullying which is easy to cross. The line between strengthening ties or creating microcultures, on the one hand, and spreading rumours, discrediting or insulting those who are the target of the gossip on the other is an extremely fine one and can be tilted towards what we know as cyberaggression [13], as this study has shown. The absence of non-verbal communication and personal contact that occurs on Internet, which is so important to ensure the quality of social relationships [7], can lead to a lack of moral sensitivity, which results in young people attacking their peers without feel any remorse, thus, rupturing the harmony of the social ecosystem and online coexistence [45].

The second hypothesis suggested that certain cognitive disengagement mechanisms could be responsible for the transition from cybergossip to cyberaggression. This hypothesis was only partially confirmed in this study. The immorality of aggressive online behaviour, shown in different meta-analyses [20,21], made us predict that the mechanisms of moral disengagement, in particular cognitive reconstruction, dehumanization and the distorting of harmful consequences for the victim, would have a mediating effect, as previously observed in previous studies [30]. This was confirmed in the case of cognitive restructuring, as it seems that both directly (offline) [46], and through digital communication (online), the mechanism of cognitive restructuring, or personal interpretation of the information issued or received, does act as a mediating process. However, in this longitudinal study, this mediating effect was not found in relation to mechanisms of moral disengagement, such as dehumanization or distortion of consequences for the victim, as was also found in some cross-sectional and correlational studies [30]. It is clear, then, that cognitive restructuring is the only mechanism that mediates the relationship between cybergossip and cyberaggression. This may be because accessing the Internet on digital devices constitutes a more impersonal context that leads adolescents to activate a mechanism which is cognitively ‘colder’, such as the restructuring of the information itself by focusing on a simple, straightforward justification of their behaviour. Meanwhile, distortion of the consequences requires more complex arguments and dehumanization of the victim also seems to require more complex semantic resources.

This study has certain limitations that must be taken into consideration when interpreting the results and the scope of the discussion. Despite the fact that it is a three-wave longitudinal study, these cover the period of a single year, so subsequent studies could be designed with wider-spaced intervals, which could perhaps give the findings greater prediction strength. In addition, this work was carried out using self-reports, which, despite being the most widely-used instruments in this field [47], can be affected by response bias or social convenience, so future studies should include multiple informants or qualitative studies. Also, it would be better to use cross-cultural studies, which would allow us to take into account the cultural variable when interpreting the results [48].

## 5. Conclusions

This work presents important findings, which put into context the importance of gossip as a scenario for consolidating the structure of the peer network in its communication and interpersonal ties, and highlights the fragility of its contents and the vulnerability faced by these peers as protagonists. This work also gives greater insight into the practical implications that can be taken into consideration for programs encouraging appropriate online behaviour, the quality of interactions and communications on social media, as well as the prevention of online bullying [49]. For instance, it is clear that ethical standards in online behaviour and autonomous, critical thinking in the face of cyberaggression or the abuse of power need to be encouraged. Although it is not universal, cyberaggression can occur frequently in the online networks of relationships built among young people and adolescents, leading to a social imbalance, which has negative implications for young people’s moral and emotional development [45]. The very fine communicative and intentional lines which separate cybergossip and cyberaggression, combined with the more impersonal, detached context of non-verbal communication and the use of arguments which are easily manipulated to create lies and ‘fake news’, seem to facilitate cyberaggression. For these reasons, we would encourage schools to try to foster moral sensitivity and humanization, as well as self-generated responsibility and critical thinking, when it comes to both posting and interpreting online information written about others. In short, the educational and socializing processes carried out in schools need to encourage young people to form an ethical code of conduct which they comply with both face-to-face and online; but for this, it is necessary that these programs are supported by families. These should be a good example in the transmission of ethics and civic values that favour the common good. Previous studies have shown how families can also become morally disengagement [50], influencing this in the emission of cyber-aggressive behaviours by young people. The union between families and the school must be essential in promoting social and moral responsibility for the actions that young people commit through the screen. For this reason, we consider it important that these programs promote joint work between family and school to promote moral development through dialogue and the work of MD mechanisms.

Finally, the harmful consequences that these practices have on the mental health of the young people involved and that has been evidenced in other works [51], should arouse the interest of health professionals and researchers to promote, together with families and professionals of the education, health programs that allow addressing this problem from a multidisciplinary perspective [52]. In particular, health promotion and prevention programs can be designed both for young people who begin to spread rumours about their colleagues and for the victims of them, since this work has shown how they can end up in cyberaggression.

## Figures and Tables

**Figure 1 ijerph-18-01000-f001:**
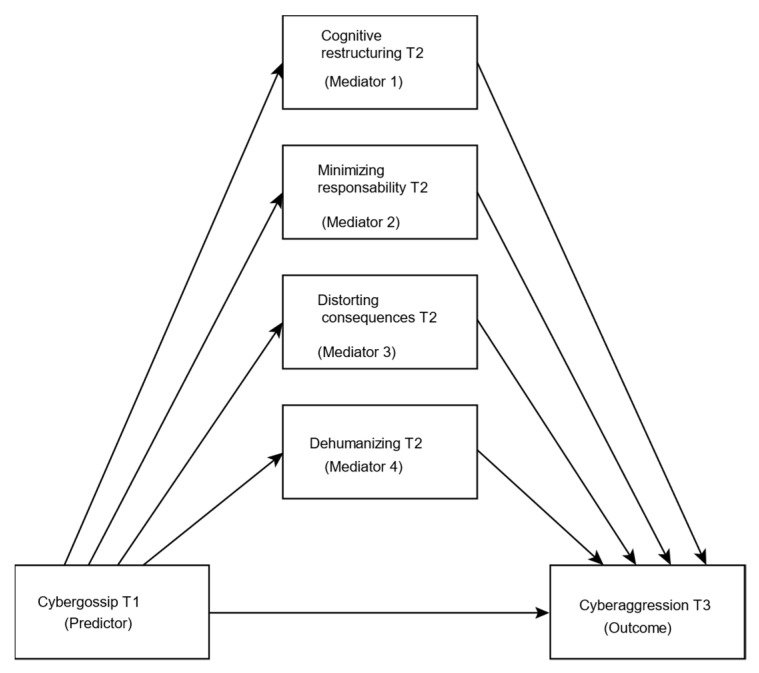
Proposed mediation model.

**Figure 2 ijerph-18-01000-f002:**
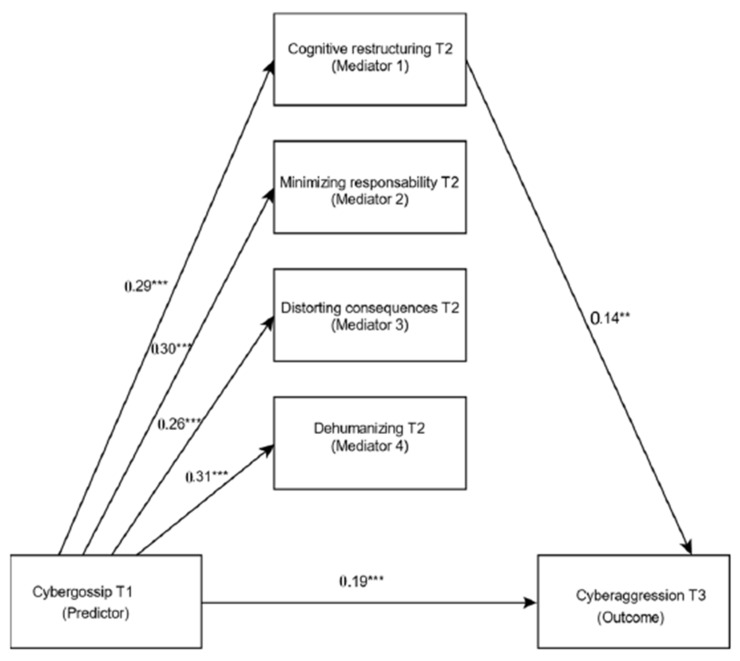
Results in the pathways of the model. Note: ** *p* < 0.01; *** *p* < 0.001.

**Table 1 ijerph-18-01000-t001:** Means, Standard Deviations, and differences by gender for all variables.

	Sample	Boys		Girls				
	M	DT	M	DT	M	SD	*t*	*d*
Cybergossip T1	0.40	0.51	0.41	0.55	0.39	0.48	0.54	0.03
Cyberaggression T1	0.10	0.26	0.12	0.32	0.08	0.19	2.60 **	0.15
RC T1	1.52	0.59	1.67	0.67	1.37	0.46	9.46 ***	0.52
MR T1	1.91	0.82	1.98	0.87	1.84	0.75	3.15 **	0.18
DC T1	1.38	0.61	1.49	0.69	1.28	0.49	6.49 ***	0.35
DH T1	1.46	0.58	1.58	0.67	1.34	0.45	7.48 ***	0.41
Cybergossip T2	0.38	0.52	0.34	0.52	0.41	0.51	−2.10 *	0.12
Cyberaggression T2	0.12	0.33	0.15	0.39	0.10	0.26	2.60 **	0.15
RC T2	1.53	0.61	1.67	0.66	1.41	0.53	7.72 ***	0.43
MR T2	1.79	0.73	1.86	0.77	1.73	0.69	3.04 **	0.17
DC T2	1.38	0.60	1.47	0.69	1.29	0.48	5.58 ***	0.31
DH T2	1.48	0.62	1.57	0.69	1.38	0.52	5.60 ***	0.31
Cybergossip T3	0.34	0.47	0.30	0.48	0.37	0.46	−2.96 **	0.16
Cyberaggression T3	0.10	0.31	0.12	0.37	0.09	0.22	1.83	0.10
RC T3	1.50	0.59	1.65	0.66	1.36	0.47	8.84 ***	0.50
MR T3	1.72	0.73	1.76	0.76	1.68	0.69	2.19 *	0.12
DC T3	1.35	0.58	1.45	0.67	1.26	0.46	6.03 ***	0.34
DH T3	1.45	0.59	1.56	0.69	1.34	0.46	6.91 ***	0.39

Note: M = mean; SD = Standard deviation; *t* = Student’s t; *d* = Cohen’s d; CR = Cognitive restructuring; MR = Minimizing responsibility; DC = Distorting consequences; DH = Dehumanizing; * *p* < 0.05; ** *p* < 0.01; *** *p* < 0.001.

**Table 2 ijerph-18-01000-t002:** Testing the mediation effect of cybergossip T1 on cyberaggression T3.

	Model 1		Model 2								Model 3	
	CA (T3)		CR (T2)		MC (T2)		DC (T2)		DH (T2)		CA (T3)	
	β	*t*	β	*t*	β	*t*	β	*t*	β	*t*	β	*t*
Age	0.03	0.90	0.15 ***	4.42	0.07	1.89	0.12 **	3.48	0.07	1.75	0.01	0.16
Gender	−0.05	−0.89	−0.35 ***	−6.84	−0.15 **	−2.76	−0.23 ***	−4.53	−0.28 ***	−5.03	0.01	0.23
CA T1	0.30 ***	6.46	0.20 ***	4.48	0.07	1.42	0.06 ***	1.42	0.24 ***	4.94	0.27 ***	5.74
CG (T1)	0.14 ***	4.13	0.24 ***	7.25	0.29 ***	8.05	0.22 ***	6.77	0.25 ***	6.99	0.09 **	2.63
CR (T2)											0.15 **	3.31
MC (T2)											0.04	1.12
DC (T2)											0.01	0.31
DH (T2)											−0.004	−0.09
*R* ^2^	0.09 ***		0.17 ***		0.10 ***		0.10 ***		0.14 ***		0.12 ***	
F	27.37		52.74		28.32		27.91		41.89		18.02	

Note: R^2^ = R^2^ adjusted; β *=* standardized regression coefficients; *t* = Student’s t; CG = Cybergossip; CA = Cyberaggression; CR = Cognitive restructuring; MR = Minimizing responsibility; DC = Distorting consequences; DH = Dehumanizing; Pearson’s correlation coefficient. ** *p* < 0.01; *** *p* < 0.001. Gender was dummy coded such that 1 = boy, 2 = girl.

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
