# Peer review of "Mechanisms of Moral Disengagement in the Transition from Cybergossip to Cyberaggression: A Longitudinal Study"

_ijerph, 2021, doi:10.3390/ijerph18031000_

Round 1

Reviewer 1 Report

I would suggest the authors to avoid excesive self-citations. Many other studies have been conducted (separately or jointly) on cyber gossip, cyberbullying and moral disengagement that has not been reviewed. I understand that authors have a long and strong line of research but the literature review in the paper seems to be biased by self-citations.

Please, briefly summarize grounds for each hypothesis.

There is insufficient description of the sample. Author/s should include a discussion of the desired sample based on a power analysis, then the procedure used (i.e., who was contacted about participation), and finally the number of participants who were involved in the study. Given the sensitivity of the questions how was anonymity and confidentiality conveyed and ensured? How long did the data collection process take in each wave?

Please, include number of approval of the Ethics committe for the current study.

Conclusion should be reviewed considering the question: What is the take-home message of this paper for health researchers and health professionals? Health variables have been not included in the study, so authors should include a rationales for the inclusion of the current study in a journal direct to health policies. Also, some conclusive remarks may be specified by the authors for the general readership

Reviewer 2 Report

Reviewer Comments

This paper addresses the important topic of cyberbullying – specifically, the paper focuses on the relationship between cybergossip and cyberbullying and examines moral disengagement as a mediator between the two.  Results of a mediational analysis revealed a direct relationship between cybergossip at T1 and cyberbullying at T2, as well as an indirect (mediation) effect through one type of moral disengagement, cognitive restructuring. Although I believe the study is intended to shed light on cybergossip as a “gateway” to cyberbullying through the process of moral disengagement, there are some significant limitations in the analysis that warrant attention before the paper is suitable for publication. In particular, although the authors make a case for gender/sex differences throughout the paper, they control for gender/sex in the analyses rather than looking at gender/sex as a moderating variable. It is quite possible that the indirect relationship between cybergossip and cyberbullying may be different for males and females. There are additional limitations that dampen enthusiasm for publication in its current form.

Abstract

Line 13 – “interviewed” is not an accurate description of survey administration.

Introduction

General comment - a definition of cybergossip and cyberbullying should be presented earlier – on p. 2 line 84 the distinction is made as “intention to do harm” – this should be clarified earlier in the introduction. Also, are cybergossip and cyberbullying on a continuum – that is, the same construct but one is more severe than the other? Or are they different in some fundamental way? Just a different magnitude of the same behavior or a different behavior?

p. 1 lines 36-38 – this sentence does not make sense – cybergossip does not necessarily imply making negative comments about others?

p. 2 lines 49-52 – the statement about gender differences is unclear- girls were involved more often in “this type of behavior” – is that cybergossip and cyberbullying? And then boys and girls use the internet and digital devices the same amount – that does not address differences in cybergossip even though they both use devices, so unclear how the second part of the sentence relates – maybe this: “Although the use of internet and digital devices is equally common among male and female youth, some research suggests that girls are more involved in cybergossip and cyberbullying than boys.”

p. 2 lines 56-60 – need citations

p. 2 lines 71-72 – regarding online attacks “…boys are initially those who do it the most, although eh balance tends to even out with age” – this seems to contradict gender difference above with girls being more involved in cybergossip and cyberbullying.

p. 4 line 141 – what is model 4?

p. 4 lines 142 to 148 – the constructs in the model do not match those in the text.

Participants

Why are there 7th graders at wave 3?

Instruments

General – I would recommend putting the cybergossip measure before the cyberbullying measure as it is first in the model

General – are the “w” statistics supposed to be alphas? If so, they should be denoted with a

p. 1 line 163 - cyberbaggression should be cyberbullying

Procedures

General – I would like to see more on recruitment – the response rate is high for studies with parental consent procedures. Also, what did the non-participants do during data collection?

Data Analysis/Results

Gender/sex differences are discussed throughout the paper and gender/sex is controlled for in the analyses – but given the emphasis made on gender/sex difference, the model really needs to include gender/sex as a moderator – it is possible that the mechanism of progressing from cybergossip to cyberbullying is different for males and females.

Cyberbullying and the 4 MD variables at T1 need to be in the model as a control variables.

Table 1 really just needs correlations for T1.  Same for the Table in the Appendix – just T1 – and it should be moved to the text, not presented as an Appendix.

There should be a figure with the results – with the pathways that are significant etc.

Discussion

p. 9 line 282 – what hypothesis – start with hypothesis 1 to parallel structure of next paragraph

p. 9 line 284 to 285 – only 3 of the MD constructs are mentioned – this happens throughout the paper

p. 9 lines 309-318 – I am not convinced that this explanation makes sense – it seems dehumanization, in particular, would be more likely to happen online. Also, if gender/sex moderates the indirect paths, this discussion will need to be revised.

p. 9 lines 324-325 – “most widely used instruments in this field” needs a citation

Other comments –

  • Sex and gender are used interchangeably through the paper – this should be cleaned up
  • Cyberbullying and cyberaggression are used interchangeably through the paper – this should be cleaned up
  • There are some grammatical issues and typos that need to be cleaned up throughout the paper
  • The tables are not formatted well – alignment issues etc.

Reviewer 3 Report

The present manuscript analyzes the relations between cybergossip and cyberbullying, and the mediating role of the moral disengagement. It is an interesting and novel study focused on topics of social relevance. The previous literature review is adequate, the design is appropriate, and the results are interesting. Its longitudinal design is also a strength of this study.

This manuscript, in my opinion, may be published with some minor changes.

Introduction

The main variables are adequately defined, and the objectives of this study are clearly justified. However, some questions need to be revised by authors:

Line 5: authors wrote “offered by new Information and Communication Technologies (ICT)”. Nevertheless, currently the Information and Communication Technologies (ICT) can no longer be considered "new", because they have already been part of the interpersonal relationships of adolescents and young people since a considerable number of years. It is advisable to delete the word "new".

Lines 144-146: “H2: Cognitive restructuring, distortion of consequences and dehumanization (T2) will be the mechanisms that mediate the relationship between cybergossip (T1) and cyberbullying (T3).” Please, review this hypothesis, considering Figure 1. In Figure 1, there is no “cyberbullying (T3)” and mediating variables are not indicated in T2 (they are indicated in T1). Moreover, “Minimizing responsibility T1” is not included in H2, but it is considered as mediating variable in Figure 1. It is necessary a clear correspondence between the two hypotheses and the proposed mediation model in Figure 1. It is also necessary to review H1 (cyberbullying is indicated in T2 in figure 1).

Methods

Participants:

T1 and T2 (cyberaggression) are included in Figure 1. However, a description of samples in T1, T2 and T3 is included in the Participants section. An exact correspondence with Figure 1 is necessary.

Authors should provide more information about participants in this study: Are these samples representative of students in their region or country? What country are samples from? What are the characteristics of the schools these students attend? Are they rural or urban schools? public or private schools? elementary schools? what number of schools participated in this study?

Instruments:

Line 174: “nine items (ie “I talk to my group of friends…”)” In this line, “ie” should be “eg”

Results

In Table 1: “DT” should be “SD”

Discussion

It would be convenient to detail in a more specific way the practical applications of these results for developing prevention programs to reduce cyberbullying. Authors may explain more about the role of schools and families in these intervention programs, considering the influence of mechanisms of moral disengagement.

Round 2

Reviewer 1 Report

The current version is better described in terms of background and methodology. Conclusions are directly derived from results and authors have corrected excesive self-citations. The paper could be published in the current state.

Reviewer 2 Report

I believe the comments have been addressed and the paper is much improved.